# Ultrafast relaxation of photoexcited superfluid He nanodroplets

M. Mudrich [1]*, A.C. LaForge [2,3], A. Ciavardini[4,17], P. O'Keeffe[4], C. Callegari [5], M. Coreno[4], A. Demidovich[5], M. Devetta [6,18], M. Di Fraia [5], M. Drabbels [7], P. Finetti[5], O. Gessner [8], C. Grazioli [19], A. Hernando[9,10], D.M. Neumark [8,11], Y. Ovcharenko[12,20], P. Piseri[6], O. Plekan[5], K.C. Prince[5], R. Richter [5], M.P. Ziemkiewicz[8,11], T. Möller[12], J. Eloranta[13], M. Pi [14,15], M. Barranco [14,15,16] & F. Stienkemeier [2]

The relaxation of photoexcited nanosystems is a fundamental process of light–matter interaction. Depending on the couplings of the internal degrees of freedom, relaxation can be ultrafast, converting electronic energy in a few fs, or slow, if the energy is trapped in a metastable state that decouples from its environment. Here, we study helium nanodroplets excited resonantly by femtosecond extreme-ultraviolet (XUV) pulses from a seeded free-electron laser. Despite their superfluid nature, we find that helium nanodroplets in the lowest electronically excited states undergo ultrafast relaxation. By comparing experimental photoelectron spectra with time-dependent density functional theory simulations, we unravel the full relaxation pathway: Following an ultrafast interband transition, a void nanometer-sized bubble forms around the localized excitation (He*) within 1 ps. Subsequently, the bubble collapses and releases metastable He* at the droplet surface. This study highlights the high level of detail achievable in probing the photodynamics of nanosystems using tunable XUV pulses.

[1] Department of Physics and Astronomy, Aarhus University, Aarhus C 8000, Denmark. [2] Institute of Physics, University of Freiburg, Freiburg im Breisgau 79104, Germany. [3] Department of Physics, University of Connecticut, Storrs, CT 06269, USA. [4] CNR-ISM, Area della Ricerca di Roma 1, Monterotondo Scalo 00015, Italy. [5] Elettra – Sincrotrone Trieste S.C.p.A., Basovizza, Trieste 34149, Italy. [6] Dipartimento di Fisica, Università degli Studi di Milano, Milan 20133, Italy. [7] Laboratory of Molecular Nanodynamics, Ecole Polytechnique Fédérale de Lausanne (EPFL), Lausanne 1015, Switzerland. [8] Chemical Sciences Division, Lawrence Berkeley National Laboratory, Berkeley, CA 94720, USA. [9] Kido Dynamics, EPFL Innovation Park Bat. C, 1015 Lausanne, Switzerland. [10] IFISC (CSIC-UIB), Instituto de Fisica Interdisciplinar y Sistemas Complejos, Campus Universitat de les Illes Balears, 07122 Palma de Mallorca, Spain. [11] Department of Chemistry, University of California, Berkeley, CA 94720, USA. [12] Institut für Optik und Atomare Physik, TU-Berlin 10623, Germany. [13] Department of Chemistry and Biochemistry, California State University at Northridge, Northridge, CA 91330, USA. [14] Departament FQA, Facultat de Física, Universitat de Barcelona, Barcelona 08028, Spain. [15] Institute of Nanoscience and Nanotechnology (IN2UB), Universitat de Barcelona, Barcelona 08028, Spain. [16] Laboratoire des Collisions, Agrégats, Réactivité, IRSAMC, UMR 5589, CNRS et Université Paul Sabatier-Toulouse 3, Toulouse Cedex 09, 31062, France. [17] Present address: CERIC-ERIC Basovizza, Trieste 34149, Italy. [18] Present address: CNR-IFN, Milano 20133, Italy. [19] Present address: CNR-IOM, Istituto Officina dei Materiali, Area Science Park - Basovizza, Trieste 34149, Italy. [20] Present address: European XFEL, Schenefeld 22869, Germany. *email: mudrich@phys.au.dk

Understanding the ultrafast response of condensed phase nanosystems to photoexcitation is essential for many research areas, including atmospheric science[1], radiation damage in biological matter[2,3], light-harvesting mechanisms in natural and artificial complexes[4,5], and photocatalysis[6]. However, the complex couplings of electronic and translational degrees of freedom often present major theoretical challenges[7]. In addition, the complexity of heterogeneous solid or liquid systems, as well as difficulties in preparing well-controlled samples and performing reproducible measurements, make it difficult to unravel the elementary steps in the relaxation process. In this respect, superfluid He nanodroplets are ideal model systems for studying the photodynamics in weakly-bound nanostructures, both experimentally and theoretically; He atoms have a simple electronic structure, interatomic binding is extremely weak, and the structure of He nanodroplets is homogeneous and nearly size-independent due to their superfluid nature[8,9]. Exploring transient phenomena associated with superfluidity is a particularly intriguing aspect of He nanodroplet spectroscopy[10,11]. By probing the dynamics of laser-excited molecular systems coupled to He droplets, one gains insight into the fluid dynamics, dissipation, and transport properties of a superfluid on the molecular scale[12–14].

The properties of pure He droplets can be directly studied using electron bombardment or XUV radiation. From previous theoretical[15,16] and static photoexcitation studies[17–23], the following dynamical response to resonant absorption of an XUV photon has been inferred: The electronic excitation created in the droplet localizes on an atomic or molecular center $He_n^*$, $n = 1, 2, \ldots$, within a few 100 fs[24]. Subsequently, a void cavity or bubble forms around $He_n^*$ due to Pauli repulsion between the excited electron and the surrounding ground state He atoms[21], which expands up to a radius of 6.4 Å[25] within about 350 fs[26]. Depending on how close to the droplet surface the excitation localizes, the bubble either collapses before fully forming thereby ejecting $He^*$ or $He_2^*$ out of the droplet, or remains in a metastable state inside the droplet[21]. Using laser-based high-harmonic light sources[27], various ultrafast processes initiated by exciting high-lying states in the autoionization continuum of He nanodroplets have been revealed, including the emission of slow electrons[22], the ejection of Rydberg atoms and excimers[28,29], and ultrafast interband relaxation[23]. However, the dynamics of low-lying states below the autoionization threshold and in particular the bubble formation have not been probed for pure He nanodroplets, neither at the strongest absorption band associated with the atomic $He^*$ $1s2p$ $^1P$ state (photon energy around $h\nu = 21.6$ eV[18]), nor at the lowest optically accessible $1s2s$ $^1S$ state ($h\nu = 21.0$ eV[18]).

In the present study, we excite these lowest excited states to directly probe the relaxation dynamics of neutral pure He nanodroplets. The experiment is carried out using tunable XUV pulses generated by the seeded free-electron laser (FEL) FERMI[30]. The comparison of time-resolved photoelectron spectra (PES) with time-dependent density functional theory (TD-DFT) calculations reveals an ultrafast three-step relaxation process. Despite the extremely weak binding of the He atoms in the droplets and the superfluid nature thereof, energy dissipation is very efficient even for the lowest excited states; more than 1 eV of electron energy is dissipated in <1 ps due to the coupling of electronic and nanofluid nuclear degrees of freedom.

## Results

### Resonant two-photon ionization scheme.
The pump-probe scheme is sketched in Fig. 1. The gray shaded area in Fig. 1a shows the absorption spectrum of He nanodroplets taken from Joppien et al.[18]; for reference, the $He^*$ atomic levels are given on the right-hand side of Fig. 1a. The massive broadening and shifting of the excited state is due to a repulsive interaction

between the $2p$ Rydberg electron and the $1s$ core electrons at large interatomic distances[31]. The straight vertical arrows illustrate the pump (red) and probe (blue) steps, realized by one XUV pulse and one time-delayed UV pulse. The electron kinetic energy, $T_e$, measured by means of electron velocity-map imaging (VMI)[30,32], is represented as a black double-sided arrow. The most likely relaxation pathway for $1s2p$ $^1P$-excited He nanodroplets is indicated by the dotted curved arrows. The inset shows a schematic view of a He nanodroplet exposed to a pair of laser pulses, containing a localized excitation marked by (∗).

### Time-resolved photoelectron spectra.
Examples of time-dependent PES measured by exciting He droplets to the $1s2s$ $^1S$ state ($h\nu = 21.0$ eV) and on the blue edge of the $1s2p$ $^1P$ band ($h\nu = 22.2$ eV) are shown in Fig. 2a and b, respectively. The horizontal dashed lines indicate the electron energy one would expect for direct $1+1'$ ionization of He by absorption of one pump and one probe photon,

$$T_e^{\text{direct}} = h\nu + h\nu' - E_i, \qquad (1)$$

where $E_i = 24.6$ eV is the ionization energy of He and $h\nu' = 4.8$ eV is held fixed. The panels on the right-hand sides show the PES at selected pump-probe delays. For positive delays (XUV first, UV second), the PES mainly consist of two spectral components in both cases a) and b). A broad feature labeled 'D' dominates the PES at short delays $t \lesssim 0.5$ ps, whereas a sharp peak 'A' becomes prominent at longer delays. The PES for each value of the pump-probe delay were fit with the sum of three Gaussian functions. The fit parameters were mildly constrained to ensure the convergence of two Gaussians to peaks 'A' and 'D', whereas the third Gaussian approximates the electron background signal. Fig. 2c and d show the amplitudes and center positions of these two peaks obtained from the fits of the PES measured at various $h\nu$. Peak D (solid lines in Fig. 2c) rises within the first 0.5 ps delay time and then slowly decreases, accompanied by a rapid increase of peak A (dashed lines). The opposite trends of these two components indicates a redistribution of population from D to A within 0.5–2.5 ps.

The energy of peak D (Fig. 2d) rapidly decreases within $t < 1$ ps, followed by a slow decrease beyond 2.5 ps. Peak A slightly shifts from 0.9 to 0.8 eV within $t < 1$ ps and remains constant thereafter. Its linewidth is limited by the resolution of the spectrometer. The final peak position matches the electron energy expected for ionization of a He atom in the lowest excited singlet state,

$$T_e^{S,\text{atom}} = E(1s2s{}^1S) + h\nu' - E_i = 0.8 \text{ eV}, \qquad (2)$$

where $E(1s2s\ ^1S) = 20.6$ eV. Therefore, we associate peak A with the ionization of a $1s2s$ $^1S$-excited $He^*$ which is either weakly bound to the droplet surface or ejected into vacuum. This interpretation is supported by PES measured for various He droplet sizes presented in the Supplementary Fig. 1 and discussed in Supplementary Note 1. While for larger droplets peak A appears slightly later and remains less intense in proportion to peak D, its position converges to the same final value (0.8 eV). Consequently, peak D is assigned to a $He^*$ located further inside the He droplet such that it is energetically shifted up. This assignment is backed by the evolution of the $He^+$ and $He_2^+$ ion yields as a function of delay, see Supplementary Fig. 2. When exciting the He droplet to its $1s2s$ $^1S$ state at $h\nu = 21.0$ eV (Fig. 2a), the initial position of peak D (1.2 eV) matches the electron energy one expects based on the droplet absorption spectrum (Fig. 1a),

$$T_e^{S,\text{drop}} = 21 \text{ eV} + h\nu' - E_i = 1.3 \text{ eV}. \qquad (3)$$

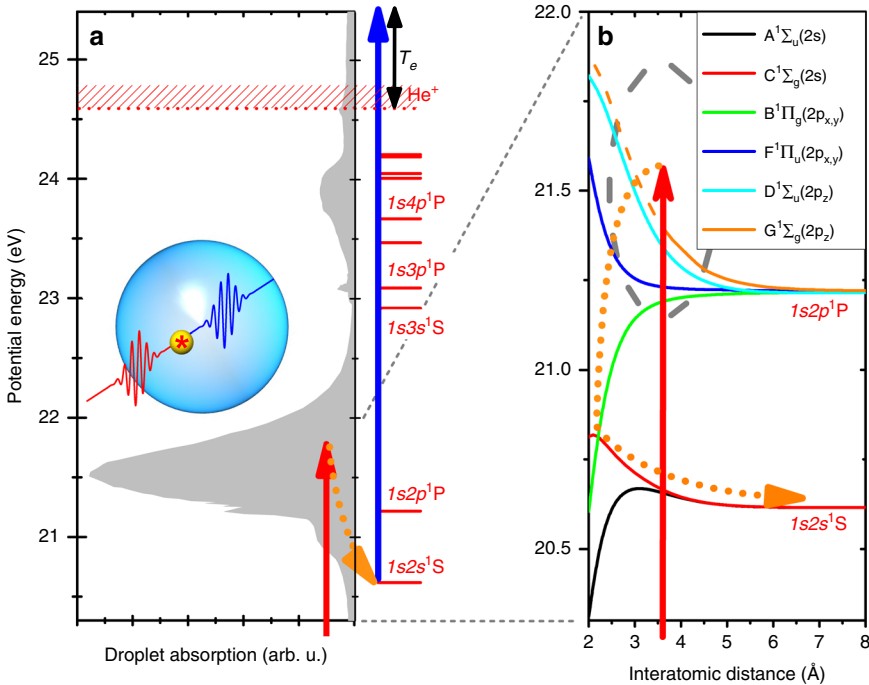

**Fig. 1 Illustration of the pump-probe scheme. a** The filled area represents the absorption spectrum of He nanodroplets taken from Joppien et al.[18]. He atomic levels are shown on the right-hand side. **b** Potential curves of the singlet-excited $He_2^*$ dimer correlating to the 1s2s $^1S$ and 1s2p $^1P$ atomic states were computed as detailed in the Methods section. The vertical straight arrows indicate the pump and probe laser pulses, the dotted curved arrows indicate the droplet relaxation pathway. The double-sided arrow in (**a**) illustrates the electron kinetic energy $T_e$.

At higher $h\nu$, where mainly the 1s2s $^1P$ droplet state is excited (Fig. 2b), feature D corresponds to a superposition of $^1S$ and $^1P$ states which relaxes into the $^1S$ droplet state faster than the cross correlation of the two laser pulses (250 fs FWHM) and thus cannot be fully resolved. Note that not all droplets evolve into the atomic $^1S$ state (peak A); in the final stage of relaxation, the state that converges to an energy 0.1–0.2 eV above the $^1S$ atomic value (feature D) and the atomic $^1S$ state are nearly equally populated.

**Potential energy curves of the $He_2^*$ excimer**. How can the extremely weakly bound, ultracold van der Waals He clusters induce ultrafast energy relaxation by up to 1.6 eV within 1 ps? To answer this question, we first consider the potential curves of the $He_2^*$ excimer correlated to the atomic 1s2s $^1S$ and 1s2p $^1P$ levels as the simplest model system for the excited He droplet, shown in Fig. 1b. The blue-shifted absorption profiles with respect to the atomic levels can be related to the steep upwards bending of the optically active A, D, and F states in the range of most probable interatomic distances (3.6 Å), indicated by the gray oval line. When exciting He in a nanodroplet, these diatomic states are expected to be strongly coupled to form a band-like structure. Following ultrafast intraband relaxation to the lowest state B of this manifold, the system further relaxes by internal conversion via the crossing of potential curves B and C at short interatomic distance, as illustrated by the dotted arrows. Subsequently, the local environment rearranges to accommodate the newly formed 1s2s $^1S$ He atom. On the longer timescale of the fluorescence lifetime, some of the He* stabilize by forming $He_2^*$ excimers[19,20,33].

**Time-dependent density functional theory calculations**. To simulate this process for He droplets in three dimensions, we carried out TD-DFT calculations for a He* excitation in the 1s2s $^1S$ state, as outlined in the Methods section. Note that this transition is forbidden in free atoms. Therefore it preferentially

takes place in the surface region of the droplets where the radially-varying He density breaks the symmetry of the free He atom and makes the transition partly allowed (see Methods).

As seen in Fig. 3, the system evolves differently depending on the initial position $d$ of He* with respect to the droplet surface. The radius of the droplet containing $N = 1000$ He atoms is $R_D = 2.2$ nm. Shown are snapshots of the He density distribution at fixed times $t$ after He* excitation. Such snapshots for intermediate values of $d$ are shown in Supplementary Fig. 3 and discussed in Supplementary Note 2. When He* is initially placed at the surface of the droplet ($d = 0$, left column), the surrounding region is locally compressed and forms a spherical dimple, while He* flies off within $t \lesssim 1$ ps. This scenario resembles the dynamics of excited alkali metal atoms which initially reside in dimple states at the droplet surface[34–37]. When He* is initially placed deeper in the bulk of the droplet ($d = 0.7$ nm, right column), first a bubble forms around He*, which then bursts at $t \approx 4$ ps, thereby allowing He* to escape out of the droplet. This scenario has also been studied theoretically and experimentally for photoexcited silver atoms[38] and indium atoms embedded in He nanodroplets[14].

Besides visualizing the dynamics ensuing excitation of the droplet, the TD-DFT model allows us to simulate the time-dependent PES, see Methods section. Figure 4a shows the resulting electron kinetic energies $T_e^{sim}$ for different values of $d$. In the case He* is initialized close to the droplet surface ($d = 0$ and 0.2 nm), $T_e^{sim}$ rapidly drops from about 1.4 eV at $t = 0$ to the final value of 0.8 eV within $t = 250–500$ fs due to prompt desorption of He*. When He* is placed deeper inside the droplet ($d = 0.4$, 0.7, and 1.2 nm), an initial fast drop of $T_e^{sim}$ from 1.6 to 0.9–1.1 eV is followed by a slow decrease to 0.9 eV at $t = 2$ ps.

The average of all curves in Fig. 4a, weighted by the geometric factor $(R_D - d)^2$, is shown in Fig. 4b as a dashed line. It nicely follows the experimental curve for the droplet feature D (red solid line in Fig. 4b) up to about 2 ps delay and eventually converges to

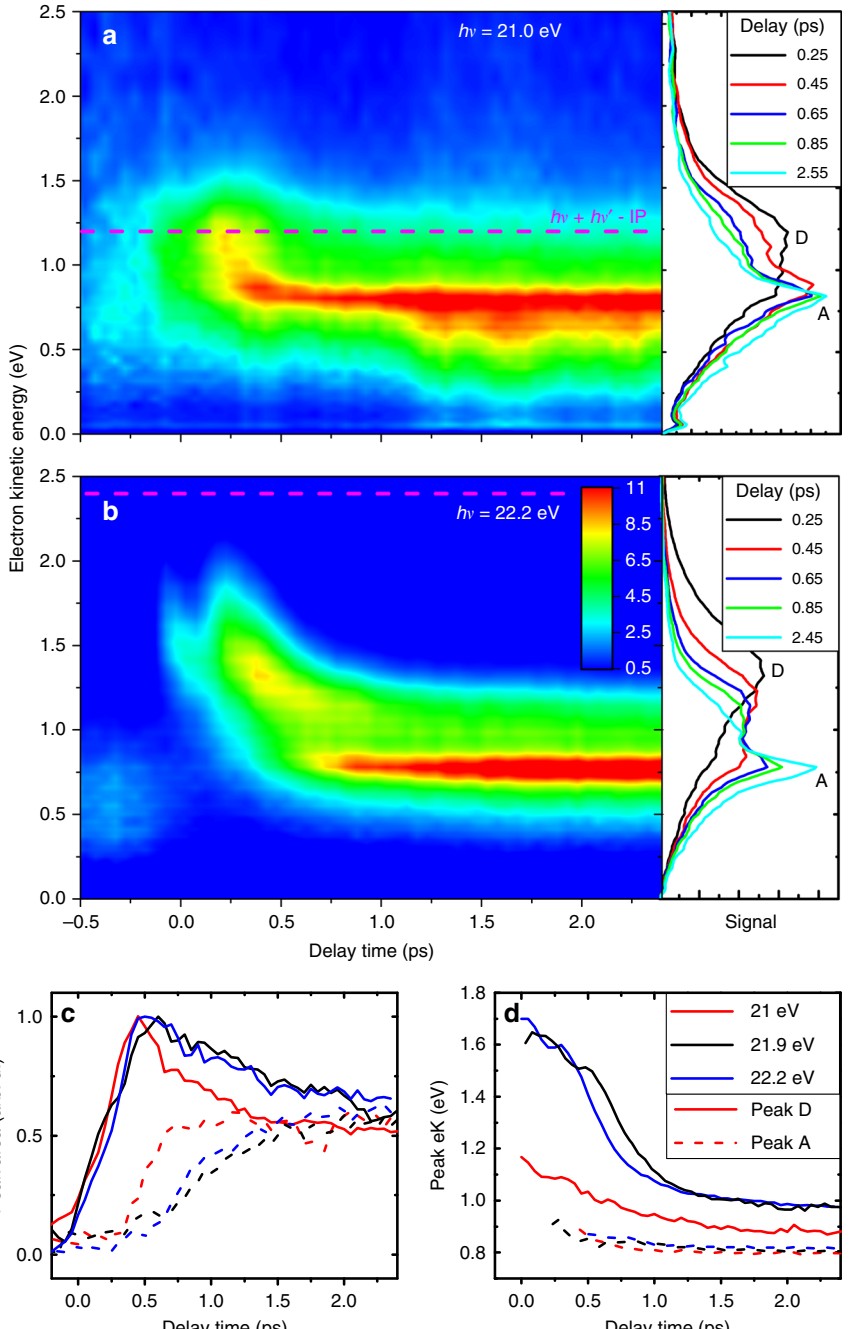

**Fig. 2 Time-resolved photoelectron spectra.** The He nanodroplets contained on average $\overline{N} = 5 \times 10^5$ He atoms and were excited to their 1s2s state at the pump photon energy $h\nu = 21.0$ eV (**a**) and their 1s2p state at $h\nu = 22.2$ eV (**b**). The probe photon energy is $h\nu' = 4.8$ eV. The horizontal dashed lines indicate the electron energy corresponding to direct two-photon ionization of He. The panels on the right-hand sides show the electron spectra recorded at various fixed pump-probe delays. The bottom panels show the results of fitting the spectra with multiple peaks (area in (**c**), position in (**d**)).

the final value of the atomic peak A. In particular, the fast drop between 0 and 1 ps coincides with the drop of peak D energy in the experimental PES (Fig. 2d) and with the appearance of peak A as the bubble forms around He*. Thereafter it slowly decreases from 0.9 to 0.8 eV as the bubble migrates to the droplet surface and eventually releases an unperturbed He*. Note that the simulated curve for $d = 0.7$ nm shows an oscillatory behavior between $t = 0.4$ and 2 ps. We attribute this nearly periodic modulation of $T_e$ to the oscillation of the He bubble around He*. He bubble oscillations around impurity atoms (Ag and In) have also been discussed[14,38].

## Discussion

From the comparison of the experimental and theoretical results, we can now map out the full picture of the relaxation dynamics of excited He nanodroplets: Initiated by the excitation of the $1s2p$ $^1P$ nanodroplet state, which is likely delocalized over several He atoms[24], ultrafast interband relaxation to the $1s2s$ $^1S$ droplet state occurs within <250 fs induced by curve crossings of the He$_2^*$ potentials (step 1). This is in line with earlier photoluminescence studies which showed that the $1s2p$ $^1P$ droplet state mainly decays by XUV-photon emission of He$_2^*$ in its A state correlating to the $1s2s$ $^1S$ state of He*[19].

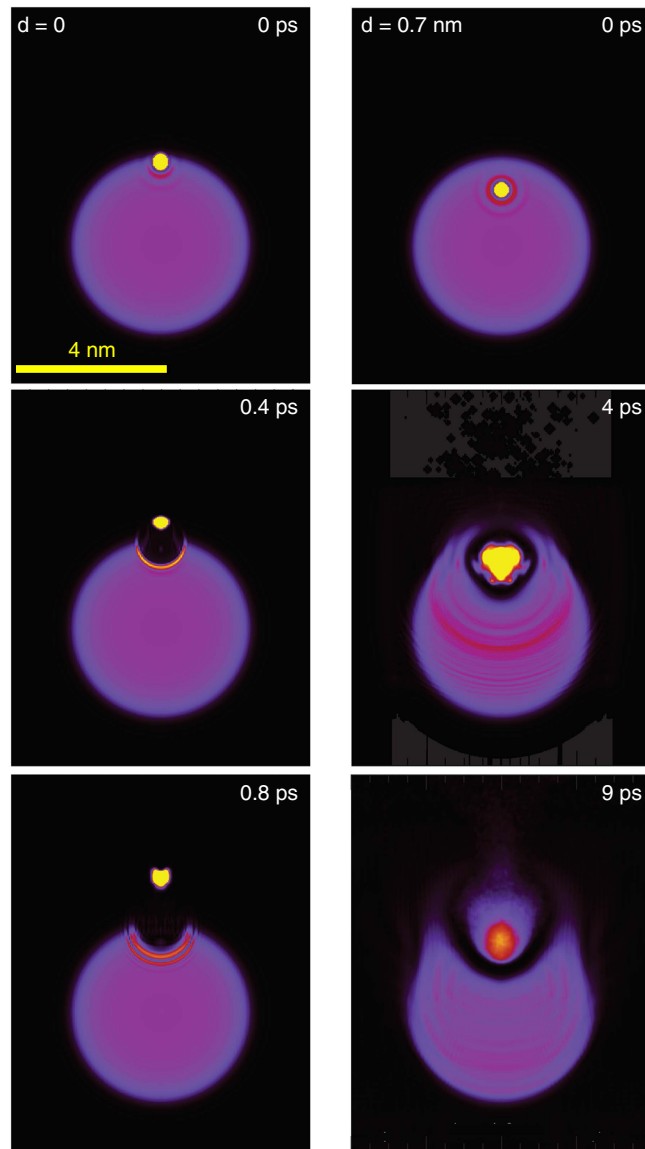

**Fig. 3 Evolution of the He density distribution.** In this simulation, the probability distribution of He* is represented as a yellow dot. The initial position of He* is 0 (left column) and at 0.7 nm (right column) away from the droplet surface.

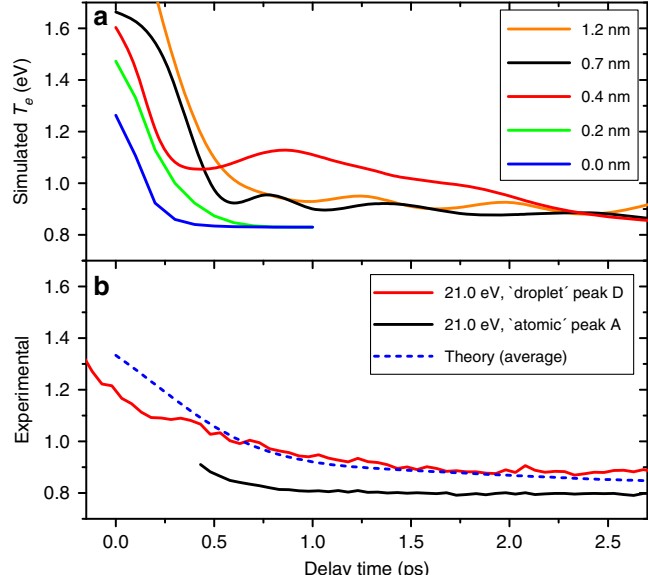

**Fig. 4** Simulated (**a**) and measured (**b**) electron energies. The He droplets were excited to the $1s2s$ state at $h\nu = 21.0$ eV. The dashed line in (**b**) is the average of the simulated curves in (**a**) for different initial positions of the He* excitation. The geometric weight of each curve as well as the experimental pulse cross correlation function are taken into account.

Further relaxation proceeds within the $1s2s\ ^1S$ droplet band by localization of the excitation due to the local opening of a void bubble around an excited He* atom (step 2). The relaxation time associated with this step (0.5 ps) is in good agreement with the established model of bubble formation around an electron, if we assume a final bubble radius of 6.4 Å[25,26]. It is nearly independent of the location of He* within the droplet and of the droplet size $\overline{N}$. This explains the weak variation of the experimental pump-probe PES when changing $\overline{N}$.

Subsequently, the bubble migrates to the droplet surface and bursts to release a free He* (step 3). The fact that in our experiment, both free and bubble-bound He* are measured at $t = 2.5$ ps shows that the migration of the bubble to the surface is a slow process that strongly depends on the initial He* location and therefore on $\overline{N}$. A recent study of the excited state dynamics of xenon clusters revealed electronic relaxation and the emission of free xenon atoms[39]. Thus, our findings appear not to be specific to He nanodroplets but of rather general relevance for weakly bound condensed phase systems. Eventually, the He* atoms that

remain attached to the droplet surface further relax by forming He$_2^*$ as seen in time-independent measurements[19,20]. The latter radiatively decay to the ground state after undergoing vibrational relaxation and partly detaching from the droplet[33].

The presented measurements show that it is now possible to follow the relaxation dynamics of free nanodroplets in great detail using ultrashort tunable XUV pulses. Diffractive imaging of He droplets and embedded impurities has recently attracted considerable attention[40–42]. However, direct time resolved imaging of the bubble dynamics is at the present stage challenging, given the small size of the bubbles and the expected low contrast. Further development of ultra-bright X-ray light sources is needed to enter the regime of atomic resolution in single clusters and will then deliver a wealth of detailed information. The present technique could be used for probing the photodynamics of more complex natural or synthetic nanosystems in various regimes of excitation of the valence shell[43] and even inner shells[44].

## Methods

The experiments described were performed at the low density matter (LDM) end station of the seeded free-electron laser FERMI[30].

**He droplet generation**. The He nanodroplets were formed by expanding He gas from a high pressure reservoir (50 bar) through a pulsed, cryogenically cooled Even-Lavie nozzle at a pulse repetition rate of 10 Hz[45]. The mean size of the He droplets formed in this way was controlled by changing the temperature of the nozzle in the range of 5–28 K.

**Light sources**. Linearly polarized XUV pulses in the photon energy range 21.0–22.2 eV were provided by the FERMI free electron laser set to the 5th harmonic of the seed laser wavelength[46]. The XUV pulses generated in this way have a bandwidth <0.1 eV and a temporal duration of about 100 fs FWHM. A Kirkpatrick-Baez mirror system was used to focus the FEL light to a spot size of 0.5 mm in the interaction region of the spectrometer. To minimize non-linear effects due to absorption of more than one photon per droplet the XUV pulses were strongly attenuated by a combination of a N$_2$ filled gas cell and an aluminum filter. The pulse energy in the interaction region was estimated to be 6 μJ.

The UV probe pulses (170 fs duration, 7 μJ pulse energy) were generated by frequency tripling part of the 775 nm Ti:Sa laser used to generate the seed light for the FEL. The UV pulses were focused to the same focal spot size as the XUV beam and superimposed with the XUV pulses in a quasi collinear geometry via reflection from a holey mirror. The temporal cross-correlation function was measured using

two-photon ionization of He atoms via the He $1s5p$ $^1P$ state. A Gaussian fit yields a FWHM of 250 fs.

**Electron detection, data acquisition, and analysis.** PES from the He nano-droplets are recorded using a VMI spectrometer, in which electrons are accelerated by electrostatic optics imaged onto a position sensitive detector consisting of a 75 mm microchannel plate and phosphor screen assembly. For each step of the pump-probe delay of 50 fs, VMI spectrometer images from 2000 shots were saved. A background subtraction procedure was implemented in which the bunches of He nanodroplets were periodically desynchronized from the FEL pulses so that spurious signals such as scattered light could be subtracted. The VMI spectrometer images for each delay were then summed and subsequently inverted using the pBasex routine to extract the photoelectron kinetic energy and angular distributions[47].

**Ab initio calculations of He–He\* and He–He⁺ potentials and transition dipole moment.** The He\*–He interaction potentials corresponding to $2s$ and $2p$ He atomic asymptotes were obtained at the CC3-EOM level[48,49] by using the Psi4 code[50]. The basis set was taken from ref. [51]. All the calculated potentials were corrected for basis set superposition errors by the counterpoise method of Boys and Bernardi[52].

The transition dipole $\overrightarrow{\mu}_{2s}$ as a function of He\*($2s$)–He($1s$) distance was evaluated at the multi-reference configuration interaction (MRCI) level using the Molpro code[53,54]. The active space consisted of the molecular states originating from $1s$ and $2s$ atomic states. These calculations employed the basis set given in refs. [55,56]. The transition dipole induced by the inhomogeneous He density in the droplet surface region is calculated as the vector sum of dipole moments of a single He\*–He pair weighted by the radial He density distribution,

$$
\begin{aligned}
\overrightarrow{\mu}_{2s}^{\text{drop}} &= \int d\mathbf{r}\, \rho(\mathbf{r})\, \overrightarrow{\mu}_{2s}(|\mathbf{r} - \mathbf{r}_X|) \\
&= \int d\mathbf{r}\, \rho(\mathbf{r})\, \big|\overrightarrow{\mu}_{2s}(|\mathbf{r} - \mathbf{r}_X|)\big| \frac{\mathbf{r} - \mathbf{r}_X}{|\mathbf{r} - \mathbf{r}_X|}.
\end{aligned}
\tag{4}
$$

We find the transition dipole moment to be peaked nearly at the He droplet radius $r_0 N^{1/3}$, $r_0 = 2.22$ Å, where it takes the value $|\overrightarrow{\mu}_{2s}^{\text{drop}}| = 0.17$ Debye.

**Time-dependent density function theory.** The dynamics of the excited He droplet was simulated using time-dependent density functional theory (TD-DFT) for droplets consisting of 1000 He atoms[15,16], to which the dynamics of the He\* atom is self-consistently coupled.

Due to the light mass of the He\* "impurity", its dynamics is followed by solving the Schrödinger equation for it, where the potential term is given by the He\*-droplet interaction. The expected high velocity of the impurity makes it advantageous to use the test-particles method for solving the Schrödinger equation[16,34]. We obtain the excess energy transfered to the photoelectron as $T_e(t) = h\nu' - [U^+(t) - U^*(t)]$. Here, the interaction energies of He\* with its local environment in the He droplet in the ($t$-dependent) initial state, $U^*(t)$ is computed as

$$
U^*(t) = \int\int d\mathbf{r}\, d\mathbf{r}'\, \Phi^2(\mathbf{r}', t)\, \rho(\mathbf{r}, t)\, \mathcal{V}_{\text{He–He}^*}(|\mathbf{r}' - \mathbf{r}|),
\tag{5}
$$

where $\Phi^2$ is the probability density of He\*, $\rho$ is the ground-state He density, and $\mathcal{V}_{\text{He–He}^*}$ is the He–He\* interaction pair potential, respectively. The interaction energy of He⁺ with the droplet in the final state, $U^+(t)$, is obtained in the same way using the He–He⁺ interaction potential, $\mathcal{V}_{\text{He–He}^+}$.

## Data availability
The data that support the findings of this study are available from the corresponding author upon request.

## Code availability
The results of the simulation were obtained by adapting to the test particles approach the ⁴He-DFT BCN-TLS computing package which is freely available at M. Pi, F. Ancilotto, F. Coppens, N. Halberstadt, A. Hernando, A. Leal, D. Mateo, R. Mayol and M. Barranco, ⁴He-DFT BCN-TLS: A Computer Package for Simulating Structural Properties and Dynamics of Doped Liquid Helium-4 Systems, https://github.com/bcntls2016/.

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

## Acknowledgements

Funding from the Deutsche Forschungsgemeinschaft (MU 2347/8-1, STI 125/19-1, and the priority program 1840 QUTIF, the Carlsberg Foundation, National Science Foundation (DMR-1828019), Carl-Zeiss-Stiftung, Grant No. FIS2017-87801-P (AEI/FEDER, UE), and Swiss National Science Foundation (200020_162434) is gratefully acknowledged. O.G., D.M.N., and M.P.Z. were supported by the U.S. Department of Energy, Office of Basic Energy Sciences, Chemical Sciences, Geosciences and Biosciences Division, through Contract No. DE-AC02-05CH11231.

## Author contributions

M.M. and F.S. conceived the experiment. C.C., M.C., A.D., M.D., M.D.F., C.G., and O.P. implemented the experiment. M.M., A.C.L., A.C., P.O'K., C.C., M.C., M.D.F., P.F., Y.O., P.P., O.P., K.C.P., R.R., M.P.Z., and F.S. performed the measurements. M.M., A.C.L., A.C., and P.O'K. analyzed the experimental data. J.E. performed the ab initio calculations. A.H., M.P., and M.B. performed the TDDFT calculations. A.C.L., P.O'K., M.D., O.G., D.M.N., T.M., J.E., M.B., and F.S. contributed to the interpretation of the data. M.M. wrote the paper with input from all other authors.

## Competing interests

The authors declare no competing interests.
