## [Peer Review File · Nature Communications]

Reviewers' comments:

Reviewer #1 (Remarks to the Author):

The paper by Mudrich et al. describes the experimental observation and modeling of ultrafast dynamics of He* in nanodroplets. The helium is excited by VUV pulses in the range of 21-22 eV, which are generated by the free electron laser FERMI. The dynamics of excited He* is probed by UV pulses impinging the target after a delay, which are creating electrons. The electron kinetic energy is analyzed as a function of pump-probe delay for excitation photon energies of 21.0, 21.9 and 22.2 eV.

The authors identify two features (D and A) in the photoelectrons spectra under all excitation conditions. At 22.2 eV, the initial photoelectron energy in maximum D is higher compared to the lower excitation photon energies. When analyzing the change in photoelectron spectra, Mudrich et al. find a fast change in the mean energy of peak D of more than 1 eV in one picosecond. Peak A shows a comparably mild energy loss rate. Based on these features and the excitation energy dependence of increasing loss with increasing excitation energy, peak D is attributed to excited He* atoms within the He-nanodroplet. Feature A is attributed to excited helium atoms in the surface region of the droplet. Hence the authors pose the question how ultrafast energy relaxation can occur in weakly bound van der Waals systems like He nanodroplets. They answer the question with simulations based on TD-DFT, showing how He* forms a bubble, how it reaches the surface if buried inside, or how it is expelled from the surface if excited on the surface. The geometry images are accompanied by simulated electron spectra that generally show a loss of kinetic energy as the geometrical relaxation of He* inside the nanodroplet occurs. The simulated spectra and the experimental ones are very similar and thus make a strong statement about the de-excitation mechanism.

As He nanodroplets are interesting systems to observe all kinds of quantum phenomena (for instance recently quantized vortices), the current study is therefore of interest to a rather large fundamental quantum physics community. The relaxation mechanisms for buried and surface excitations are known from previous experiments on buried silver and indium on one hand, and on alkali atoms staying on the surface on the other hand. The new aspect here is that pure He nanodroplets can be used to observe relaxation phenomena when exciting He* using VUV pulses. This, done in a time-resolved fashion, is fundamentally new and will for sure excite the readers of Nature Communications. The data are presented in a clear way, the argumentation is logical and can be followed throughout the paper. I have however some concerns on the simulation and the conclusions drawn from them: I am left with the impression that the surface excitations leave the cluster in the simulation on a timescale less than a picosecond. This would result in atomic He*, which would show extremely narrow electron peaks. However, the data show rather broad features in peak A. What is that due to? Is the simulation overestimating the effect of desorption or is the experimental linewidth limited by other effects?

If everything is in good agreement with the past impurity measurements of silver, indium and alkalis, are there any aspects that are different? As He* is not a metal, I guess one could argue there should be differences and those will need to be discussed.

In addition, I find a couple of smaller points that ne

ed to be addressed:

- Fig. 1: color coding into blue and purple does not work as the two colors are too similar
- Page 2: what is meant by 'unfavorable' in the Rydberg interaction?
- Fig. 2: the color scheme in part c and d is hard to decipher, I guess it is meant that A is always dashed and D always solid, however this is only shown in the color of the 21 eV excitation
- Page 3, discussion of fig. 2: the main text should contain information on how peaks D and A are distinguished and how the energy is fitted (what is the lineshape?)
- Page 3, second last paragraph: 'but nearly the same fraction remain...' – same fraction with respect

to what?

- Page 3, last paragraph: 'ultrafast internal conversion proceeds due to level crossings' – I cannot identify the crossings in Fig. 1 and also the dashed path is not showing any crossing on the way, opposite to the claim in the text
- Fig. 4: it should be mentioned in the main text, what the sampling was from different depth in order to generate the theory average
- Page 5, last paragraph: ...'in which electrons are accelerated by electrostatic imaged': seems there is a word missing

Reviewer #2 (Remarks to the Author):

In their manuscript entitled "Ultrafast relaxation of photoexcited superfluid He nanodroplets", Mudrich et al. present a study of the relaxation pathway of superfluid helium nanodroplets that were resonantly excited by tunable femtosecond extreme-ultraviolet (XUV) pulses from a seeded free-electron laser. The authors compare experimental photoelectron spectra with time-dependent density functional theory simulations.

Ultrafast relaxation of photon-excited systems is of fundamental importance for a wide range of applications. For example, the detailed relaxation pathway of biological molecules excited by radiation emitted by x-ray free-electron lasers determines what kind of radiation damage occurs during molecular imaging, and with that what the possible resolution for structure determination is. Many attempts have been made to model these processes, but it would be very exciting to actually validate these simulations.

Mudrich et al. present a remarkable study of the details of the relaxation pathway, and the consistency with results from time-dependent density functional theory are very reassuring. However, it is not immediately clear how well this kind of technique would perform on more complex systems. Also, the nano voids have not been observed directly, and it is conceivable that diffractive imaging could be used in a pump-probe setting to directly follow the void evolution. The authors should have a brief discussion on both these issues in the paper.

The technical content of the paper appears sound and complete.

I recommend publication in Nature Communications.

Reply letter to the Reviewers

We thank both Reviewers #1 and #2 for their thorough proof reading of our manuscript, for their constructive criticism, and overall positive assessment of our work. Their critiques and corrections were very helpful for further improving our manuscript, which we hope is now acceptable for publication. The changed text passages in the manuscript are marked in red.

Reviewer #1 (Remarks to the Author):

The paper by Mudrich et al. describes the experimental observation and modeling of ultrafast dynamics of He* in nanodroplets. The helium is excited by VUV pulses in the range of 21-22 eV, which are generated by the free electron laser FERMI. The dynamics of excited He* is probed by UV pulses impinging the target after a delay, which are creating electrons. The electron kinetic energy is analyzed as a function of pump-probe delay for excitation photon energies of 21.0, 21.9 and 22.2 eV.

The authors identify two features (D and A) in the photoelectrons spectra under all excitation conditions. At 22.2 eV, the initial photoelectron energy in maximum D is higher compared to the lower excitation photon energies. When analyzing the change in photoelectron spectra, Mudrich et al. find a fast change in the mean energy of peak D of more than 1 eV in one picosecond. Peak A shows a comparably mild energy loss rate. Based on these features and the excitation energy dependence of increasing loss with increasing excitation energy, peak D is attributed to excited He* atoms within the He-nanodroplet. Feature A is attributed to excited helium atoms in the surface region of the droplet.

Hence the authors pose the question how ultrafast energy relaxation can occur in weakly bound van der Waals systems like He nanodroplets. They answer the question with simulations based on TD-DFT, showing how He* forms a bubble, how it reaches the surface if buried inside, or how it is expelled from the surface if excited on the surface. The geometry images are accompanied by simulated electron spectra that generally show a loss of kinetic energy as the geometrical relaxation of He* inside the nanodroplet occurs. The simulated spectra and the experimental ones are very similar and thus make a strong statement about the de-excitation mechanism.

As He nanodroplets are interesting systems to observe all kinds of quantum phenomena (for instance recently quantized vortices), the current study is therefore of interest to a rather large fundamental quantum physics community. The relaxation mechanisms for buried and surface excitations are known from previous experiments on buried silver and indium on one hand, and on alkali atoms staying on the surface on the other hand. The new aspect here is that pure He nanodroplets can be used to observe relaxation phenomena when exciting He* using VUV pulses. This, done in a time-resolved fashion, is fundamentally new and will for sure excite the readers of Nature Communications.

The data are presented in a clear way, the argumentation is logical and can be followed throughout the paper. I have however some concerns on the simulation and the conclusions drawn from them: I am left with the impression that the surface excitations leave the cluster in the simulation on a timescale less than a picosecond. This would result in atomic He*, which would show extremely narrow electron peaks. However, the data show rather broad features in peak A. What is that due to?

Is the simulation overestimating the effect of desorption or is the experimental linewidth limited by other effects?

The experimental linewidth is entirely limited by the resolution of the VMI spectrometer. The figure below shows an electron spectrum (black line) recorded using a free atomic He jet which is photoionized by XUV pulses at a photon energy of 25.3 eV. This spectrum was taken for calibration purposes during the same beamtime as the He droplet spectra presented in the manuscript. For comparison, the electron spectrum shown in Fig. 2 b) of the manuscript, which was recorded by pump-probe ionization of He nanodroplets at photon energies 22.2 + 4.8 eV for a delay of 2.45 ps, is also included. The widths of the sharp peak 'A' and that of the photoline from free He atoms are nearly identical. This confirms that, within the resolution of the VMI spectrometer, peak 'A' is associated with $1s2s^1S$ -excited He* atoms which are either weakly bound to the droplet surface or ejected into vacuum.

To make this point clear we add a sentence to the text on p. 3: “Its linewidth is limited by the resolution of the spectrometer.”

If everything is in good agreement with the past impurity measurements of silver, indium and alkalis, are there any aspects that are different? As He* is not a metal, I guess one could argue there should be differences and those will need to be discussed.

The fundamental difference between the previously studied dynamics of impurities and the present study of pure He nanodroplets is that here, the initial excitation can be delocalized over several He atoms in the droplet, as it was theoretically found for small excited He clusters in Ref. [24]. This ultrafast electron motion may allow the excitation to roam about the He nanodroplet during a few hundred fs before localizing by forming a bubble around it. To probe this ultrafast electronic dynamics experimentally, one would have to repeat this experiment using shorter laser pulses, ideally in the sub-fs range, which is unfortunately not yet possible with the present lasers.

This electron motion is not included in the simulations presented in this work. This may explain the slight deviation between the experimental and the theoretical results (Fig. 4 b) at short delays < 0.5 ps. However, the overall good agreement between theory and experiment indicates that the dynamics is mostly determined by the bubble formation, which is similar for the cases of electronically excited impurities.

To address this point in more detail, we modify the paragraph on p. 5 to “From the comparison of the experimental and theoretical results we can now map out the full picture of the relaxation dynamics of excited He nanodroplets: Initiated by the excitation of the $1s2p\ ^1P$ nanodroplet state which is likely delocalized over several He atoms [24], ultrafast interband relaxation to the $1s2s\ ^1S$ droplet state occurs within < 250 fs induced by curve crossings of the He_2^* potentials (step 1). This is in line with earlier photoluminescence studies which showed that the $1s2p\ ^1P$ droplet state mainly decays by XUV-photon emission of He_2^* in its A state correlating to the $1s2s\ ^1S$ state of He^* [19].

Further relaxation proceeds within the $1s2s\ ^1S$ droplet band by localization of the excitation due to the local opening of a void bubble around an excited He^* atom (step 2).”

In addition, I find a couple of smaller points that need to be addressed:

- Fig. 1: color coding into blue and purple does not work as the two colors are too similar

The colors of the two vertical arrows were changed to pink and blue.

- Page 2: what is meant by ‘unfavorable’ in the Rydberg interaction?

As detailed in Ref. [31], the excited-state He_2^* potential curves are repulsive at large R (internuclear distance) but exhibit an attractive well at small R . The repulsion at large R results from an unfavorable exchange interaction of a Rydberg orbital mostly on one He with the two core orbitals on the other He. We suggest to change the sentence to:

“The massive broadening and shifting of the excited state is due to a repulsive interaction between the $2p$ Rydberg electron and the $1s$ core electrons at large interatomic distances [31].”

- Fig. 2: the color scheme in part c and d is hard to decipher, I guess it is meant that A is always dashed and D always solid, however this is only shown in the color of the 21 eV excitation

The legends of panels c) and d) have now been changed such that every line color/style in these panels is explicitly specified.

- Page 3, discussion of fig. 2: the main text should contain information on how peaks D and A are distinguished and how the energy is fitted (what is the lineshape?)

We have inserted in the main text the sentences:

“The PES for each value of the pump-probe delay were fit with the sum of 3 Gaussian functions. The fit parameters were mildly constrained to ensure the convergence of two Gaussians to peaks ‘A’ and ‘D’, whereas the third Gaussian models the electron background signal.”

The sentences “The PES for each value of the pump-probe delay were fit with a constrained 3 Gaussian fit. The time variation of the resulting fit parameters reveal the temporal behavior of the various ionization channels.” are removed from the “Methods” section.

- Page 3, second last paragraph: ‘but nearly the same fraction remain...’ – same fraction with respect to what?

What was meant is that in the final stage of relaxation, peaks A and D have the same peak area. We change the sentence to: “Note that not all droplets evolve into the atomic 1S state (peak A), but in the final stage of relaxation, the state that converges to an energy 0.1-0.2 eV above the 1S atomic value (feature D) and the atomic 1S state are nearly equally populated.”

- Page 3, last paragraph: ‘ultrafast internal conversion proceeds due to level crossings’ – I cannot identify the crossings in Fig. 1 and also the dashed path is not showing any crossing on the way, opposite to the claim in the text

Fig. 1 is modified to better visualize the potential energy curve structure correlating to the $1s2p^1P$ excited state of He. The potential energy curve of state G is now drawn in the full shown range of R. A gray oval line is added which indicates the range of R and energies where we expect the diatomic states to be strongly coupled to form a band-like structure when exciting He in a droplet. Following ultrafast intraband relaxation to the lowest state B of this manifold, intraband relaxation proceeds via the crossing of potential curves B and C. This relaxation path is illustrated by the dotted arrows in Fig. 1 b).

To make this point clear, the paragraph on p. 4 is modified to “The blue-shifted absorption profiles with respect to the atomic levels can be related to the steep upwards bending of the optically active A, D and F states in the range of most probable interatomic distances (3.6 \AA), indicated by the gray oval line. When exciting He in a nanodroplet, these diatomic states are expected to be strongly coupled to form a band-like structure. Following ultrafast intraband relaxation to the lowest state B of this manifold, the system further relaxes by internal conversion via the crossing of potential curves B and C at short interatomic distance, as illustrated by the orange dotted arrow.”

- Fig. 4: it should be mentioned in the main text, what the sampling was from different depth in order to generate the theory average

In the meanwhile, one more TDDFT simulation was carried out for the case that the He^* is initially located even deeper inside the He nanodroplet at $d=1.2 \text{ nm}$. The corresponding electron energy is

added to Fig. 4 a). The averaged simulated electron energy, shown in Fig. 4 b), is updated. The following details are added to the sentence on p. 4-5:

“The average of all curves in Fig.4 a), weighted by the geometric factor $(R_D-d)^2$, is shown in Fig. 4 b) as a dashed line.”

- Page 5, last paragraph: ...'in which electrons are accelerated by electrostatic imaged': seems there is a word missing

Yes, the word “optics” was missing, it is now inserted; thanks for pointing this out.

Reviewer #2 (Remarks to the Author):

In their manuscript entitled "Ultrafast relaxation of photoexcited superfluid He nanodroplets", Mudrich et al. present a study of the relaxation pathway of superfluid helium nanodroplets that were resonantly excited by tunable femtosecond extreme-ultraviolet (XUV) pulses from a seeded free-electron laser. The authors compare experimental photoelectron spectra with time-dependent density functional theory simulations.

Ultrafast relaxation of photon-excited systems is of fundamental importance for a wide range of applications. For example, the detailed relaxation pathway of biological molecules excited by radiation emitted by x-ray free-electron lasers determines what kind of radiation damage occurs during molecular imaging, and with that what the possible resolution for structure determination is. Many attempts have been made to model these processes, but it would be very exciting to actually validate these simulations.

Mudrich et al. present a remarkable study of the details of the relaxation pathway, and the consistency with results from time-dependent density functional theory are very reassuring. However, it is not immediately clear how well this kind of technique would perform on more complex systems. Also, the nano voids have not been observed directly, and it is conceivable that diffractive imaging could be used in a pump-probe setting to directly follow the void evolution. The authors should have a brief discussion on both these issues in the paper. The technical content of the paper appears sound and complete. I recommend publication in Nature Communications.

We suggest to modify the last paragraph of the manuscript and to include a few additional references:

“The presented measurements show that it is now possible to follow the relaxation dynamics of free nanodroplets in great detail using ultrashort tunable XUV pulses.

Diffraction imaging of He droplets and embedded impurities has recently attracted considerable attention [Gomez:2012,Langbehn:2018,Rupp:2017]. However, direct time resolved imaging of the bubble dynamics is at the present stage challenging, given the small size of the bubbles and the expected low contrast. Further development of ultra-bright x-ray light sources is needed to enter the regime of atomic resolution in single clusters and will then deliver a wealth of novel information. The present technique could be used for probing the photodynamics of more complex natural or synthetic nanosystems in various regimes of excitation of the valence shell [LaForge:2019] and even inner shells [Lackner:2016].“

Further changes to the manuscript

The list of authors was reordered and two more coauthors (O. Gessner and D. M. Neumark) were included to account for their contributions in supervising M. Ziemkiewicz and for helping interpret the experimental data.

The sentence on p. 4

“This scenario has been studied theoretically for photoexcited silver atoms [38], and experimentally for indium atoms embedded in He nanodroplets [14].” is slightly modified for the sake of being more accurate:

“This scenario has also been studied theoretically and experimentally for photoexcited silver atoms [38] and indium atoms embedded in He nanodroplets [14].”

The caption of Fig. 4 was changed to explicitly explain the various curves shown in a):

“Comparison of simulated (a) and measured (b) electron energies for droplet excitation of the 1s2s state at $h\nu=21.0$ eV. The dashed line in b) is the average of the simulated curves in (a) for different initial positions of the He* excitation. The geometric weight of each curve as well as the experimental pulse cross correlation function are taken into account.”

The sections “Acknowledgements” and “Author contributions” were previously incomplete and are now updated.

REVIEWERS' COMMENTS:

Reviewer #1 (Remarks to the Author):

I thank the authors for considering my comments. The paper is ready for publication.